Nucleotide variation and balancing selection at the Ckma gene in Atlantic cod: analysis with multiple merger coalescent models

Árnason Einar einararn@hi.is
Halldórsdóttir Katrín
Institute of Life and Environmental Sciences, University of Iceland , Reykjavík , Iceland
Ellegren Hans
Electronic publication date: 2015 Feb 24
Publication date: 2015
Volume: 3
Electronic Location ID: e786
Received 2014 Oct 9; Accepted 2015 Feb 3
Copyright: © 2015 Árnason and Halldórsdóttir
Copyright year: 2015
Copyright holder: Árnason and Halldórsdóttir
License: This is an open access article distributed under the terms of the Creative Commons Attribution License, which permits unrestricted use, distribution, reproduction and adaptation in any medium and for any purpose provided that it is properly attributed. For attribution, the original author(s), title, publication source (PeerJ) and either DOI or URL of the article must be cited.
License URL: https://creativecommons.org/licenses/by/4.0/

Keywords: Balancing selection, Multiple-merger coalescent, Atlantic cod, Time scales, Ckma

Funding: Icelandic Science Foundation 40303011 University of Iceland Research Fund Svala Árnadóttir private foundation Funding was provided by Icelandic Science Foundation grant of excellence (nr. 40303011), a University of Iceland Research Fund grant, a Svala Árnadóttir private foundation grant to Einar Árnason and a doctoral grant from the University of Iceland Research Fund to Katrín Halldórsdóttir. The funders had no role in study design, data collection and analysis, decision to publish, or preparation of the manuscript.

==============================
High-fecundity organisms, such as Atlantic cod, can withstand substantial natural selection and the entailing genetic load of replacing alleles at a number of loci due to their excess reproductive capacity. High-fecundity organisms may reproduce by sweepstakes leading to highly skewed heavy-tailed offspring distribution. Under such reproduction the Kingman coalescent of binary mergers breaks down and models of multiple merger coalescent are more appropriate. Here we study nucleotide variation at the Ckma (Creatine Kinase Muscle type A) gene in Atlantic cod. The gene shows extreme differentiation between the North (Canada, Greenland, Iceland, Norway, Barents Sea) and the South (Faroe Islands, North-, Baltic-, Celtic-, and Irish Seas) with FST > 0.8 between regions whereas neutral loci show no differentiation. This is evidence of natural selection. The protein sequence is conserved by purifying selection whereas silent and non-coding sites show extreme differentiation. The unfolded site-frequency spectrum has three modes, a mode at singleton sites and two high frequency modes at opposite frequencies representing divergent branches of the gene genealogy that is evidence for balancing selection. Analysis with multiple-merger coalescent models can account for the high frequency of singleton sites and indicate reproductive sweepstakes. Coalescent time scales vary with population size and with the inverse of variance in offspring number. Parameter estimates using multiple-merger coalescent models show that times scales are faster than under the Kingman coalescent.

Introduction

High fecundity translates into large excess reproductive capacity that would allow organisms to withstand substantial natural selection enabling them to bear the entailing high genetic load. Based on the concept of the cost of natural selection (Haldane, 1957) high-fecundity organisms relative to low-fecundity organisms should at any time be able to adapt a larger proportion of their genome to meet various environmental challenges. Trying to explain the paradox of sexual reproduction Williams (1975) in his Sex and Evolution book argues that high-fecundity coupled with heavy mortality of young (type III survivorship) may be able to pay the 50% fitness cost of meiosis. He developed several models, such as the Elm/Oyster and the Cod/Starfish models, which emphasize the importance of high-fecundity for selection. Williams also discussed the concept of reproductive sweepstakes. There is no heritability of fitness and sexual reproduction continuously assembles Sisyphean genotypes (from Sisyphus who was punished to roll a boulder up a hill only to see it roll back down, and having to repeat his actions forever). The distribution of offspring numbers is highly skewed, heavy-tailed and with high variance (lognormal). That is Williams’s fitness distribution. The environment factors are envisioned as acting in a sequence of selective filters. With only a few factors (e.g., temperature, salinity, etc.) there nevertheless can be an enormous number of different sequences of selective filters (environments) that do not recur. Hence a winning genotype is not permanent and must be continuously reassembled. Natural selection increases the variance in offspring number and thereby reduces effective population size genome-wide. Neutral variation will therefore drift faster under pervasive natural selection.

Coalescent theory (Kingman, 1982a; Kingman, 1982b) traces the genealogy of a sample and is very useful for making statistical inferences from molecular population genetic data. However, in an extreme case under a winner-take-all sweepstakes reproduction, all samples would coalesce immediately in the previous generation (Árnason, 2004) and there would be no variation. However, this extreme case is not realistic. The Kingman coalescent, which is derived from Wrigth/Fisher models of low fecundity non-skewed offspring distributions, assumes a bifurcating genealogy and is not appropriate for reproduction of this kind (Eldon & Wakeley, 2006; Schweinsberg, 2003; Wakeley, 2013; Tellier & Lemaire, 2014). Some organisms may exhibit both high fecundity and highly skewed offspring distributions. The coalescent for such organisms will lie somewhere between the extreme of a winner-take-all sweepstakes coalescent and the Kingman coalescent. For these organisms the Λ coalescent allowing multiple mergers of ancestral lineages at any one generation (Pitman, 1999; Sagitov, 1999; Donnelly & Kurtz, 1999; Eldon & Wakeley, 2006; Schweinsberg, 2003; Sargsyan & Wakeley, 2008) or Ξ coalescent allowing simultaneous multiple mergers of ancestral lineages (Schweinsberg, 2000; Möhle & Sagitov, 2001) may be more appropriate. Wakeley (2013) gives an overview of the development of coalescent theory in new directions. There is also active development of statistical inference methods associated with multiple merger coalescents (e.g., Birkner, Blath & Eldon, 2013b; Eldon et al., 2015). Studies on the high fecundity organisms Pacific oyster Crassostrea gigas (Hedgecock & Pudovkin, 2011) and Atlantic cod Gadus morhua (Linneaus, 1758) (Árnason & Pálsson, 1996; Árnason, Pálsson & Petersen, 1998; Árnason et al., 2000; Carr & Marshall, 1991a; Carr et al., 1995; Pepin & Carr, 1993; Árnason, 2004) have provided data for a number of tests of some of the new coalescent models (Eldon & Wakeley, 2006; Eldon, 2011; Eldon & Degnan, 2012; Steinrücken, Birkner & Blath, 2013; Birkner, Blath & Eldon, 2013b). A high number of singletons is a feature of sequence studies of high fecundity organisms such as the Atlantic cod. This is expected under models of multiple merger coalescents and, therefore, they perform better than the Kingman coalescent by capturing the high frequency of singletons. Atlantic cod thus provides a model for studies applying the multiple merger coalescent. In this paper we apply some of these new methods for Λ coalescents as appropriate neutral null models for high fecundity organisms in a study of balancing selection at a gene showing extreme spatial differentiation in Atlantic cod.

In general the time scale of multiple merger coalescent models can be much shorter than for a Kingman coalescent. For example, under the Beta(2 − α, α) coalescent model time scales depend on Nα−1 (Schweinsberg, 2003; Eldon et al., 2015). Under the extreme winner-take-all sweepstakes coalescent mentioned above, all individuals would be sibs differing only by new mutations. However, this is an extreme case. Real world multiple merger coalescents lie somewhere between this extreme and the Kingman coalescent. Applying multiple merger coalescents does not imply that we are sampling siblings or that samples from say Greenland and Norway share the same parents. Möhle (1998) and elsewhere shows that for large N the Kingman coalescent is robust because the influence of structure of various types (selfing, age structure, geographic structure) occurs on a shorter time scale than the time scale of the coalescent. Under the Kingman coalescent, the expected time to the most recent coalescence of a sample of n individuals is 2/(n(n − 1)) (×2Ne generations). Although the general robustness of the Kingman coalescent breaks down under multiple merger coalescents, coalescence times will nevertheless be longer than a single generation. Although time scales under multiple merger coalescents are shorter than under Kingman coalescent, and for example our estimates are square root or cube root of Ne, they are longer than a single generation of the winner-take-all sweepstakes coalescent.

A dense genomic map of genetic variation in humans (and in model organisms) allows scanning the genome for signatures of natural selection (Voight et al., 2006; Sabeti et al., 2007; Storz, 2005). The density of the genetic maps and sensitivity of the various methods used influences what percentage of the human genome we observe to show footprints of selection (Voight et al., 2006; Sabeti et al., 2007; Storz, 2005). It is safe to say that only a small percentage of single nucleotide polymorphisms (SNPs) show footprints of selection in the low fecundity humans (Akey, 2009; Pickrell et al., 2009). For microsatellite loci, 2% (13/624) were detected as outliers when African and non-African human populations were compared (Storz, Payseur & Nachman, 2004). In contrast, comparable genome level studies in Atlantic cod find that 11% (26 out of 235) of independent SNPs (Moen et al., 2008) are FST outliers (by method of Beaumont & Nichols, 1996) and 4% SNPs (70 out of 1641 Bradbury et al., 2010) are Bayscan outliers (by method of Foll & Gaggiotti, 2008) likely undergoing selection. Similarly one fourth of microsatellite loci in Atlantic cod (Nielsen, Hansen & Meldrup, 2006) are FST outliers. This supports our thesis that a considerable fraction of the Atlantic cod genome may be simultaneously under selection for different adaptations.

More than half of the 70 outliers in Bradbury et al. (2010) study of Atlantic cod show adaptive parallel clines related to temperature on both the western and eastern side of the Atlantic Ocean. They show that multiple genes, located in three independent linkage groups, are involved. There are single genes as well as blocks of genes in “genomic islands” (Bradbury et al., 2013; Hemmer-Hansen et al., 2013). Some of the genes or blocks of genes show clear spatial patterns while other genes show complex spatio-temporal patterns in contrast to no differentiation of non-outlier (neutral) loci (Poulsen et al., 2011; Therkildsen et al., 2013). For example, a locality in West Greenland shows great similarity to coastal areas in Iceland, implying either parallel adaptation on a fine scale or patterns of gene flow that are hard to reconcile with geographic distance. Another study (Hemmer-Hansen et al., 2014) adds even more complexity of population structure at outlier loci with little or no difference at non-outlier neutral loci.

A study of differentiation among four Atlantic cod populations along the coast of Norway (Moen et al., 2008) showed no differentiation among presumably neutral non-outlier loci with an average F¯ST=0.0012. In contrast, among the outlier loci, presumably under selection, the FST ranged from 0.08 to extreme differention of 0.83 with an average F¯ST=0.27. Here we analyze in detail nucleotide variation at a large fragment of the Ckma gene (encoding a muscle isoform A of creatine kinase) showing extreme spatial differentiation (Moen et al., 2008) to understand the nature of selection.

Creatine kinases (CK) are crucially important in bioenergetic processes in cells and tissues (Wallimann et al., 1992; Wallimann, Tokarska-Schlattner & Schlattner, 2011). The creatine kinase/phosphocreatine system (CK/PCr) is an intracellular energy shuttle. CK generates Phosphocreatine (PCr) at the sites of ATP production in glycolysis and oxidative phosphorilation in mitochondria and regenerates ATP from PCr at subcellular sites of ATP use by ATPases. The physiological advantage is to provide a spatial and temporal energy buffer storing and releasing energy in and from PCr. Importantly, the rate of intracellular diffusion of both Creatine (Cr) and PCr is one and three orders of magnitude faster than diffusion of ATP and ADP respectively (see Wallimann et al., 1992; Wallimann, Tokarska-Schlattner & Schlattner, 2011, for a detailed account of the CK/PCr system).

We thus have here a gene with a well defined and well understood function. The gene shows extreme spatial differentiation most likely due to selection considering the contrasting behavior of neutral non-outliers. We apply methods of multiple merger Λ coalescents, as a new and appropriate null model of neutrality for organisms with highly skewed heavy-tailed offspring distributions, to nucleotide variation of the gene to better understand the nature of selection.

Materials and Methods

Population sampling

We randomly sampled 180 individual cod from various localities from the distributional range of Atlantic cod (Fig. S1). The samples come from our large sample database of greater than 20,000 individuals. All localities are represented with at least 100 individuals (except the White Sea with 24 individuals). The localities are the waters around Newfoundland (New), Greenland (Gre), Iceland (Ice), Faroe Islands (Far), Norway (Nor), and the Barents Sea (Bar), North Sea (Nse), Celtic Sea (Cel), Irish Sea (Iri), Baltic Sea (Bal), and the White Sea (Whi). We took a large sample from Iceland and stratified the sampling to get about 8–10 individuals from the other localities to cover the widest geographic range possible with our database. After cloning, sequencing and quality checking as detailed below we had 122 individals covering a wide geographic area from the Southwest/Northwest to the Northeast and South.

We included samples of the closely related taxa Arctic cod Boreogadus saida (Lepechin, 1774) (Bsa) and Greenland cod G. ogac (Richardson, 1836) (Gog) both sampled in Greenland waters as well as Pacific cod G. macrocephalus (Tilesius, 1810) (Gma) and Walleye pollock Theragra chalcogramma (Pallas, 1811) (Gch) sampled from the Pacific ocean as outgroups. Carr et al. (1999) and Pogson & Mesa (2004) discuss the relationship and biogeography of these taxa. Coulson et al. (2006) provide the most comprehensive account based on mitochondrial genomics. They consider Arctic cod to be an outgroup for all these taxa. Atlantic cod and Walleye pollock are closely related taxa and Pacific cod slightly more distant. Pacific cod and Walleye pollock represent two separate but nearly simultaneous invasions of the Pacific. The Atlantic cod vs. Pacific cod split is dated at 4 mya and the Atlantic cod vs. Walleye pollock split is dated at 3.8 mya using conventional rates of mtDNA evolution (see time scales below). Coulson et al. (2006) suggested a nomenclature revision from Theragra chalcogramma to Gadus chalcogrammus (Pallas, 1814) for Walleye pollock that has been accepted by the American Fisheries Society (Page et al., 2013). We follow the new nomenclature hereafter. Greenland cod is a recent reinvasion of Pacific cod into the Arctic and Coulson et al. (2006) consider it to be a subspecies of Pacific cod.

The Icelandic Committee for Welfare of Experimental Animals, Chief Veterinary Office at the Ministry of Agriculture, Reykjavik, Iceland has determined that the research conducted here is not subject to the laws concerning the Welfare of Experimental Animals (The Icelandic Law on Animal Protection, Law 15/1994, last updated with Law 157/2012). DNA was isolated from tissue taken from dead fish on board research vessels. Fish were collected during the yearly surveys of the Icelandic Marine Research Institute. All research plans and sampling of fish, including the ones for the current project, have been evaluated and approved by the Marine Research Institute Board of Directors. Samples were also obtained from dead fish from marine research institutes in Norway, the Netherlands, Canada and the US that were similarly approved by the respective ethics boards. The samples from the US used in this study have been described in Cunningham et al. (2009) and the samples from Norway in Árnason & Pálsson (1996). The samples from Canada consisted of DNA isolated from the samples described in Pogson (2001). The samples from the Netherlands were obtained from the Beam-Trawl-Survey (http://www.wageningenur.nl/en/Expertise-Services/Research-Institutes/imares/Weblogs/Beam-Trawl-Survey.htm) of the Institute for Marine Resources & Ecosystem Studies (IMARES), Wageningen University, the Netherlands, which is approved by the IMARES Animal Care Committee and IMARES Board of Directors.

Molecular analysis

We used sequences associated with the Moen et al. (2008) high FST SNP’s (Gm366-0514 with an FST = 0.83, Gm366-1022 with an FST = 0.82, and Gm366-1073 with an FST = 0.82) to make probes to search our Atlantic cod BAC library (GAMH, made for us by Amplicon Express, www.amplicon-express.com). We had positive clones 454 sequenced (Microsynth, Framingham, Massachusetts, USA) and obtained a 34,223 bp scaffold containing the gene of interest. From this sequence we generated primers (Table S1) for PCR amplifying a 4,000 bp fragment for population studies. Our scaffold largely but not entirely aligned to GeneScaffold 4232 of the Atlantic cod genome sequence (Star et al., 2011) (www.ensemble.org). We confirmed our primers using the Atlantic cod genome sequence. Our BAC library was made from a single individual from Bay of Faxa (Reykjavik) Iceland and the genomic sequence is based on a specimen from the North East Arctic cod in Norway (Star et al., 2011). The conformity of primer sequences between the two specimens from widely separated geographic localities means that the primers should amplify the fragment of interest in individuals taken from widely separate geographic areas. However, it does not preclude the possibility of ascertainment bias for example due to variation in primer binding sites. The amplification primers were long (Table S1), which may facilitate annealing in spite of some variation in primer binding site. Samples from all localities were PCR amplified without issue and there were no signs of ascertainment bias in the molecular results.

Table 1 Summary statistics of polymorphism of 2,500 bp fragment of the Ckma gene, 711 bp fragment HbA2 gene and 1,021 bp fragment of the Myg gene in Atlantic cod.

Group	n	S	H	hˆ	Kˆ	θˆS	πˆ	Dˆ	
Ckma all	122	87	72	0.959	10.62	0.0067	0.0043	−1.13ns	
Ckma North	86	65	51	0.941	5.12	0.0054	0.0015	−1.97ns	
Ckma South	36	45	23	0.891	3.61	0.0045	0.0015	−2.43**	
Ckma A allele	43	49	28	0.907	4.37	0.0047	0.0018	−2.20**	
Ckma B allele	79	53	44	0.930	3.10	0.0044	0.0013	−2.33**	
HbA2 all	114	11	11	0.338	0.37	0.0030	0.0005	−2.09*	
HbA2 North	95	9	9	0.347	0.39	0.0025	0.0005	−1.95*	
HbA2 South	19	3	4	0.298	0.32	0.0016	0.0005	−0.95ns	
Myg all	45	30	24	0.901	2.74	0.0071	0.0028	−2.03*	
Myg North	36	28	20	0.894	2.65	0.0069	0.0027	−2.12*	
Myg South	9	10	7	0.944	3.22	0.0037	0.0033	−0.58ns	
Notes.

n Sample size

S number of segregating sites

H number of haplotypes

hˆ haplotype diversity

Kˆ average number of pairwise differences

θS scaled population size from S

πˆ nucleotide diversity

Dˆ Tajima’s

ns ns is not significant.

* represents P < 0.05.

** represents P < 0.01.

We Topo-TA cloned fragments into pCR XL-TOPO vector (Invitrogen, Waltham, Massachusetts, USA). We sequenced clones with M13 primers and sequencing primers (Table S1) using BigDye Terminator kit (Applied Biosystems, Waltham, Massachusetts, USA) and performed sequencing on ABI 3100 and ABI3500XL (Applied Biosystems) automated sequencers.

We applied the same methods and sequenced 711 bp of the Hemoglobin α 2 (HbA2) locus (Halldórsdóttir & Árnason, 2009a; Halldórsdóttir & Árnason, 2009b; Borza et al., 2009) and 1021 bp of the myoglobin (Myg) locus (Lurman et al., 2007). The previous studies on these genes (Halldórsdóttir & Árnason, 2009a; Halldórsdóttir & Árnason, 2009b; Borza et al., 2009; Lurman et al., 2007) had not found any signs of selection, and therefore we used them for neutral locus comparisons. The HbA2 data were of 114 Atlantic cod individuals and 13 individuals of various closely related taxa. The Myg data were from 45 Atlantic cod individuals and two individuals of Pacific cod. Other closely related taxa did not amplify for Myg. The HbA2 and Myg individuals covered much the same geographic localities as Ckma.

All sequences have been deposited in Genbank with Ckma accession numbers KM624178–KM624309, HbA2 accession numbers KM624310–KM624436, and Myg accession numbers KM624437–KM624483.

Population genetic analysis

We base called, assembled and edited sequence reads using phred, phrap and consed (Ewing et al., 1998; Ewing & Green, 1998; Gordon, Abajian & Green, 1998). We aligned sequences using muscle (Edgar, 2004), inspected alignments using seaview (version 4) (Gouy, Guindon & Gascuel, 2009) and generated maximum likelihood trees with phyml (Guindon & Gascuel, 2003) under seaview. We used R (R Core Team, 2013) and the ape, pegas, seqinr, ade4, adegenet, and LDheatmap packages (Paradis, Claude & Strimmer, 2004; Paradis, 2010; Charif & Lobry, 2007; Dray & Dufour, 2007; Jombart & Ahmed, 2011; Shin et al., 2006) and various functions written by us for managing, analyzing, and plotting the data. We used the MLHKA program (Wright & Charlesworth, 2004) for a maximum likelihood HKA test (Hudson, Kreitman & Aguadé, 1987) based on the Kingman coalescent.

By PCR amplifying and cloning of fragments, polymerase copy errors in the PCR reaction inevitably will be found in clones. The coalescent methods are especially sensitive to singleton variants and errors that would enter into the data as singleton variants should be removed. To remove PCR errors and ensure authenticity of natural variation among individuals, we sequenced three clones from each individual. We claim that taking three clones is sufficient to eliminate PCR errors among clones of an individual and yield a consensus sequence of one allele from that individual. We are taking three copies (clones) of two items (chromosomes or alleles A and a). Any two of three will always be the same allele (A and A or a and a). A third clone (order is not important) will be of that same allele with probability 1/2 and of the alternative allele from the other chromosome with probability 1/2. One of the three has probability 1/2 of being different from the two that are the same. In the first case, a consensus sequence will be a true consensus of that allele. In the second case, a consensus sequence will be a true consensus except at sites where the third clone (alternative allele) matches one of the other clones. That is when a naturally occurring site variant or a PCR error in the third clone matches a PCR error in one of the other two clones. This scenario is expected to be a rare event. The effect of such a rare event would be to generate variation that would look like recombination thus, if anything, reducing measures of linkage disequilibrium.

We thus got consensus sequences for a number of individuals. In some cases parts of a clone had low quality sequence. We visually inspected all variant sites using the above-mentioned tools. To maximize the number of individuals and the size of the sequenced fragment, we struck a balance between number of individuals and quality of sequence. We removed individuals with short sequences and removed individuals that were not covered by three clones. Also, we eliminated regions with a phred quality less than 30. We thus obtained consensus sequences of three clones from each of 122 Atlantic cod and 10 individuals of closely related taxa covering three fragments of the gene (Fig. S2) concatenated to give a total sequence of 2,500 bp.

We analyzed sequence variation for statistics of neutrality and selection using DNAsp (Rozas et al., 2003) and R functions. Site frequency spectra are a most important summary statistics for coalescent analysis of nucleotide data (Wakeley, 2009). We analyzed site frequency spectra using the Kingman coalescent (Kingman, 1982a) and statistical methods developed for multiple merger Λ coalescents (Birkner, Blath & Eldon, 2013b). We used software from Bjarki Eldon (Birkner, Blath & Eldon, 2013b) (http://page.math.tu-berlin.de/~eldon/programs.html to estimate various parameters of the multiple merger Λ coalescents. In particular, we used the minimum ℓ2 distance (Birkner, Blath & Eldon, 2013b) (sum of squares) to estimate the parameter α of the Beta(2 − α, α) coalescent (Schweinsberg, 2003) and the ψ parameter of the point-mass coalescent (Eldon & Wakeley, 2006). Using these estimates, we generated expected site-frequency spectra for the models and compared them to our observed spectra using a likelihood ratio G test with the multiple-merger coalescent models nested within the Kingman coalescent. We also used the overall ℓ2 distance (square root of ℓ2) to compare the observed and expected site frequency spectrum of the three genes, Ckma, HbA2, and Myg. We used software from Bjarki Eldon to estimate parameters of algebraic (A, γ) and exponential (E, β) growth models (Eldon et al., 2015) and compared the observed site frequency spectra for the three genes to expectations based on these growth models using the ℓ2 distance.

Results

Gene and protein

The Ckma gene encodes creatin kinase muscle isoform A (CKMA). The locus is 3604 base pairs (bp) in GeneScaffold 4232 (coordinates 332764 to 336367, gene name ENSGMOG00000008778 in the cod genome, www.ensemble.org, Star et al. (2011)). The gene has seven exons (Fig. S2). Ensemble reports 382 amino acids (aa). However, both genescan (http://genes.mit.edu/GENSCAN.html) and fgenesh (www.softberry.com) predicted 381 aa. The www.ensemble.org sequence adds a Glycine (G) residue in position 323 apparently due to incorrect splicing at the junction of the last two exons.

For mapping the gene, the SNP locus cgpGmo-S497 at position 19.5 cM (see Appendix S3 in Supplemental data of Borza et al., 2010) in linkage group CGP16 is found in a partial cDNA mRNA sequence (Genbank accession number EX184243) (Hubert et al., 2010; Borza et al., 2010) matching the Ckma gene.

There are seven paralogous genes found in the Atlantic cod genome (www.ensemble.org) encoding mitochondrial, brain and muscle isoforms of Creatine Kinase. The protein sequence of the two alleles A and B in Atlantic cod and of all the closely related taxa studied were of the CKMA isoform (Fig. S3). The variation reported is thus from orthologous genes.

Nucleotide variation and divergence

The variants of Ckma in Atlantic cod fell into two distinct and divergent groups, which we refer to as A and B alleles or haplogroups (Fig. 1 and Fig. S4). They were fixed for a C vs T at site 1,732 in the concatenated sequence (Table S2). The alleles also differed at 19 additional sites (Fig. 2 and Table S2). However, there was variation at these 19 sites that was segregating at a low frequency within one or both alleles which was evidently from recombination.

Figure 1 Maximum likelihood tree of Ckma variation (A and B alleles) among 122 individual Atlantic cod and 10 individuals of four closely related outgroup O taxa, Boreogadus saida Bsa, Gadus chalcogramma Gch, Gadus macrocephalus Gma, and Gadus ogac Gog.

Localities and color codes for Atlantic cod are the waters of Canada (Nova Scotia and Newfoundland) Can, Greenland Gre, Iceland Ice, Norway Nor, Faroe Islands Far, and from the Barents Sea Bar, White Sea Whi, North Sea Nse, Baltic Sea Bal, Celtic Sea Cel, and Irish Sea Iri.

The divergence of the A and B alleles has arisen after the speciation between Gadus morhua and its Pacific closely related species G. macrocephalus or G. chalcogrammus. The gross, DXY, and net, Da, nucleotide divergence (see also Cruickshank & Hahn, 2014) between the A and B alleles was about one half that of the divergence between the closely related taxa (Table S3). The Ckma and HbA2 divergences between the closely related taxa are very similar, but the Myg divergence is about twice that (Fig. S5 and Table S3). The variance of times to coalescence is large so it is not unexpected to find differences in divergence among genes. There is nothing in the behavior of Myg and HbA2 to indicate deviation from the multiple merger null hypothesis of neutrality. In contrast with the results of Coulson et al. (2006), the maximum likelihood tree for Ckma (Fig. 1) and divergence estimates (Table S3) imply that separation of G. chalcogrammus predates the separation of G. macrocephalus and G. morhua. Similarly, the HbA2 locus showed the same pattern that G. chalcogrammus is outside of G. macrocephalus and G. morhua (Fig. S6). Unfortunately the Myg locus did not yield sequences for G. chalcogrammus.

All summary statistics showed high variation for Ckma (Table 1). In particular, nucleotide diversity πˆ was high relative to the scaled population size θˆS, resulting in a non-significant Tajima’s Dˆ. This was due to the great number of high heterozygosity sites differing between the two alleles (Fig. 2 and Table S2). Considering the North and South population and the A and B alleles separately, there was much less variation. Although there were several polymorphic sites within both A and B alleles (Fig. 2 and Table S2), nucleotide diversity was lower than for the entire sample, and the relative difference of πˆ and θˆS for each allele was greater resulting in negative and significant Tajima’s Dˆ. The HbA2 gene had a very low haplotype and nucleotide diversity but disparity with θˆS gave overall a negative and significant Tajima’s Dˆ. In congruence with divergence measures, the Myg locus had high haplotype and nucleotide diversity, albeit lower than Ckma, but overall a negative and significant Tajima’s Dˆ.

Figure 2 Heterozygosity per nucleotide site of Ckma locus among A alleles (red A, n = 43), B alleles (blue B, n = 79), and all individuals combined (magenta C, n = 122).

Boxes represent exons, start (red), internal (magenta) and terminal (blue). Green boxes represent sequenced fragments trimmed to Phred score of at least 30. The black circles mark the three SNPs of Moen et al. (2008), Gm366-0514 locus with an FST = 0.83, Gm366-1022 locus with an FST = 0.82, and Gm366-1073 with an FST = 0.82 from left to right respectively. Crosses mark mutant sites relative to outgroup that were fixed or nearly fixed among A alleles. Triangles mark mutant sites relative to outgroup that were fixed or nearly fixed among B alleles. Gadus macrocephalus individual 152047 was used as the outgroup.

There were five non-synonymous changes segregating as singleton sites within Atlantic cod (Tables S2 and S4). Two of these were also segregating as singletons within B. saida and G. macrocephalus, and one other singleton was also found in G. macrocephalus. B. saida was fixed for a Glycine (GGT codon) for which the other taxa have a Glutamine (CAG codon) with changes in all three sites of the respective codon (aa number 242). Assuming independent mutations and depending on the path of evolution of that particular codon, all three changes may have been non-synonymous.

There was considerable linkage disequilibrium (measured as D′) throughout the gene (Figs. S7 and S8). Linkage disequilibrium measures are sensitive to allele frequency (Hedrick, 1987) and in general there is no measure that is independent of allele frequencies (Lewontin, 1988). In Fig. S8 we have therefore excluded singleton sites because they will always show maximum linkage disequilibrium. However, low frequency sites generate noise so the signal of linkage disequilibrium is hard to see. We therefore used sites with minor allele frequency greater than an arbitrary frequency of 0.1 (Fig. S7), which includes all the intermediate allele frequency (high heterozygosity) polymorphisms and gets rid of low frequency variants that generate noise in the linkage disequilibrium plots. The high frequency sites gave the clearest sign of two blocks of sites with almost full linkage disequilibrium both among sites within and between the blocks. The two blocks are separated by a site of recombination (site 691 in Table S2). On both the A and B allele backgrounds, both the ancient c major allele and the derived t minor allele at site 691 were geographically widespread (the t on an A allele background was found in individuals from the Baltic and from Iceland and the c on the B allele background was found in individuals from throughout the North ranging from the White Sea, Barents Sea, Norway, Iceland, Greenland, and Canada).

Two other sites (site three and 12 in Fig. S7 that are sites 578 and 1,444 in Table S2) showed slight reduction in linkage disequilibrium (Fig. S7) and therefore some signs of recombination. Other sites (such as sites 509, 660, 1,075 in Table S2) also showed some evidence of recombination. In all these cases, the recombinant gametic types with respect to the A and B allelic backgrounds were geographically widespread in general agreements with the result above for site 691.

The results of a maximum likelihood HKA test of selection that is based on the Kingman coalescent (Wright & Charlesworth, 2004) gave a selection parameter k = 2.12 in the direction of balancing selection (Table S5). However, the results were not statistically significant possibly because of too high variation among the presumed neutral loci (HbA2 and Myg) used for comparison in the test.

Spatial differentiation

The variation of the Ckma gene was spatially patterned. The A allele was overall at a high frequency of 97% in an area that we call South (Faroe Islands, North Sea, Baltic Sea, Celtic Sea and Irish Sea) (Table S6). Conversely, the B allele was at a high frequency of 92% in an area that we call North ranging from the Northwest (Nova Scotia and Newfoundland in Canada) through Greenland, Iceland, Norway, Barents Sea and the White Sea. There was variation among localites within each region, with some localities having zero frequency, presumably due to low sample sizes. We do not have genotypic data and cannot test for Hardy-Weinberg equilibrium. The differentiation of North and South was evident in interlocality FST values (Table S7) and an overall FST = 0.763 between North and South. There was no significant differentiation among localities within either the North or the South, but very high and significant differentiation between North and South localities. Similarly, there was great differentiation between the A and B alleles with an FST = 0.804. This was in stark contrast to the lack of differentiation between North and South at the HbA2 (FST = 0.004) and Myg (FST = − 0.029) loci.

The high differentiation was mostly due to the great number of high heterozygosity sites differing between the two alleles (Fig. 2 and Table S2). Three of the sites were the SNPs already found by Moen et al. (2008) with an FST = 0.82 for north and south localities along the coast of Norway. The high frequency sites showed indications of recombination between the A and B alleles (see for example patterns of segregating sites for individuals 105698, 124401, 105657, 200500, 118129, 119535, 118147, and 106620 in Table S2).

There were also several high heterozygosity polymorphic sites within both the A and B alleles (Fig. 2). This variation, however, did not show geographical patterns (Table S2). For example sites 1,050 and 1,428 mutated relative to outgroup within the A alleles were found among individuals from Iceland, White Sea, Celtic Sea, Faroe Islands and the Baltic. Similarly within the B alleles high heterozygosity sites 656, 691, 1,340, and 1,444, which were mutated relative to the outgroup, were all widespread among North localities ranging from the Northwest to the Northeast Atlantic (Fig. S1).

Site frequency spectra

The unfolded site frequency spectrum for the Ckma gene was trimodal (Fig. 3), with a mode at singleton sites, a mode at 43, and a mode at 79. The latter modes were at opposite frequencies out of a total of 122 and represented the A and B lineages of the genealogy. The Kingman coalescent did not fit the data well. Both the Beta(2 − α, α) and point-mass coalescent models gave a much better fit (Table S8), in particular by capturing the singleton class. None of the coalescent models captured the modes at 43 and 79.

Figure 3 Unfolded site frequency spectrum of Atlantic cod Ckma gene.

Gadus macrocephalus was used as the outgroup. Number of individuals n = 122. Theroretical expectation under Kingman coalescent (red dots), Beta(2 − α, α) coalescent (magenta squares), and point-mass coalescent (blue stars).

In contrast the site frequency spectra for the HbA2 and Myg genes were L shaped with a high peak at singleton sites (Figs. S9 and S10). Again, the Kingman coalescent did not fit well but both multiple merger coalescent models captured the high frequency of singleton sites.

The site frequency spectra of the A and B alleles alone were bimodal, with a high singleton class and peaks around 40 and 78 respectively (Fig. S11). The high frequency modes for the two alleles, at 40 and 78 respectively, resulted because most of the high frequency and high heterozygosity sites that separate the two alleles were not fixed within each allele presumably due to recombination between the alleles (Table S2, and see examples presented above).

Coalescent parameter estimates

Following Birkner, Blath & Eldon (2013b), we used the ℓ2 distance, the sum of the squared differences between the observed and expected site frequency spectrum (scaled with the number of segregating sites), for estimating parameters of two Λ coalescent models, αˆ for the Beta(2 − α, α) and ψˆ for the point-mass coalescent (Table 2, Figs. S12 and S13). The Kingman coalescent, a null model for which α = 2.0, had the highest ℓ2 indicating worst fit among the models. The HbA2 and Myg loci had an αˆ=1.00 and a ψˆ=0.23. The Ckma locus had overall a considerably higher α and lower ψ. The parameter estimates for the Ckma alleles separately were similar to those of the presumed neutral loci HbA2 and Myg.

Table 2 Parameter values minimizing the ℓ2 distance (sum of squares) between observed and expected unfolded site frequency spectra for nuclear genes and for mtDNA variation of various localities.

Based on method of Birkner, Blath & Eldon (2013b). Parameters α of the Beta(2 − α, α), and ψ of the point-mass coalescent and their respective ℓ2. The ℓ2(0) is based on the Kingman coalescent for which α = 2. For the mtDNA Carr et al. refers to Carr & Marshall (1991a), Car & Marshall (1991b); Carr et al. (1995), Pepin & Carr (1993), AP 1996 refers to Árnason & Pálsson (1996), and SA 2003 refers to Sigurgíslason & Árnason (2003).

Source	αˆ	ψˆ	ℓ2αˆ	ℓ2ψˆ	ℓ2(0)	n	Reference	
Nuclear locus	
Hba2	1.000	0.230	0.035	0.016	0.431	113	This study	
Myg	1.000	0.225	0.010	0.018	0.230	45	This study	
Ckma	1.280	0.070	0.006	0.007	0.141	122	This study	
CkmaA	1.100	0.170	0.017	0.012	0.161	43	This study	
CkmaB	1.140	0.120	0.006	0.015	0.189	79	This study	
Locality for mtDNA	
Newfoundland	1.550	0.015	0.014	0.028	0.084	378	Carret al.	
Greenland	1.945	0.005	0.072	0.071	0.072	78	Árnason et al. (2000)	
Iceland	1.550	0.010	0.006	0.050	0.078	519	Árnason et al. (2000)	
Norway	1.895	0.015	0.093	0.089	0.095	100	AP 1996	
White Sea	2.000	0.005	0.551	0.554	0.551	109	Árnason, Pálsson & Petersen (1998)	
Faroe Islands	1.555	0.050	0.059	0.055	0.093	74	SA 2003	
Baltic Sea	2.000	0.005	0.105	0.109	0.105	109	Árnason, Pálsson & Petersen (1998)	
Atlantic	1.530	0.010	0.006	0.055	0.249	1278	Árnason (2004)	

For comparison we also estimated the parameters for the entire dataset of mtDNA variation in the North Atlantic (Árnason, 2004) and the various subsamples making up that total sample using the unfolded site frequency spectrum with G. macrocephalus as the outgroup (Table 2 and Fig. S13). These have been previously analysed using the folded site frequency spectrum (see for example Birkner, Blath & Eldon, 2013b; Steinrücken, Birkner & Blath, 2013). For the total sample, spanning a similar geographic range as the nuclear genes, the parameter estimates differed from the nuclear loci with αˆ=1.53 and ψˆ=0.01. The large samples from Newfoundland and Iceland and the sample from the Faroe Islands gave similar values. The values for Greenland, Norway, White Sea, and Baltic Sea were much closer to the results for the Kingman coalescent (α = 2.0). For these localities homoplasies were more frequent in the data than for the total and the large samples. Homoplasies will reduce the number of singletons and move such sites towards the right tail of the site frequency distribution. This explains the higher values for these localities.

Models of multiple merger coalescents and population growth

It is important to see how a locus deviates from a null model of neutrality to understand selection. Here the null model is multiple merger Λ coalescents instead of the Kingman coalescent. Following Birkner, Blath & Eldon (2013b) we used the ℓ2 distance, the square root of the sum of the squared differences between the observed and expected site frequency spectrum. The overall ℓ2 distance for the three loci between the observed site frequency spectrum and expectations based on the two Λ coalescent models are in Table S9. The Ckma had the highest overall distance (the worst fit). There is clearly something special about the Ckma locus that was not seen among the other loci. In particular, the trimodal site frequency spectrum is a sign of natural selection. We did not see these for the other genes. Admittedly, this is not a formal test of selection; however, Ckma behaved differently. This is a locus specific behavior that is most likely a sign of selection.

The high frequency of singletons is predicted both by population growth and by Λ and Ξ multiple merger coalescents. Eldon et al. (2015) found that the weight of the right tail of the site frequency spectrum may have features allowing one to distinguish between population growth and Λ coalescents. Eldon et al. (2015) have developed methods for such analysis which we apply here. Using both the ℓ2 distance and approximate log likelihood, we find (Table S10) that the algebraic (A, γ) and exponential (E, β) growth models gave very similar fits for each of the three genes. Again, as with the multiple merger coalescent models (Fig. S11), the Ckma gene stood out and had the worst fit. The Myg gene showed equally good fit to the the two growth models and the Beta(2 − α, α) coalescent model. For both the Ckma and HbA2 genes, the growth models showed worse fit than the coalescent models. However, this comparison of ℓ2 distances does not constitute a formal test as stated above.

Discussion

Genes and proteins

The CKMA protein is highly conserved among the investigated taxa. The single aa difference between B. saida and the other species presumably is adaptive, with all sites of the codon having changed. The few aa variants were all singletons in the sample. In fact, most of the variation is in non-coding regions and all the high heterozygosity sites in coding regions are synonymous changes. Given the high conservation of the protein and the high variation among silent and non-coding sites that are indicative of the mutational pressure, the singleton non-synonymous changes are likely slightly deleterious and will be removed by purifying selection. Some or even all of the silent and non-coding differences between the A and B alleles may be functional control elements important in expression in different tissues or under different environments. The potential functional differences remain to be studied.

The HbA2 and Myg genes have well-defined functions. They are probably under purifying selection. They were taken as independent genes in separate linkage groups for comparison. A caveat is that genetic variation at unlinked sites may be correlated and not independent in high fecundity populations with skewed distribution of offspring (Eldon & Wakeley, 2008; Birkner, Blath & Eldon, 2013a). The question remains, however, whether and to what extent such dependence impacts inference.

Three hypotheses

We discuss three possible explanations for the observed patterns of great divergence of the A alleles and B alleles, their spatial differentiation, and the trimodal site-frequency spectrum. The first explanation is the isolation/admixture hypothesis, the second is the Ξ coalescent of simultaneous multiple mergers in any one generation, and the third is the balancing selection hypothesis. Our interpretation is that the evidence favors balancing selection.

Ancient isolation and recent admixture

First, there is the possibility of recent admixture of anciently separated and divergent gene pools that have come together in a hybrid zone of secondary contact (Bowcock et al., 1991; Bernardi, Sordino & Powers, 1993; Guinand, Lemaire & Bonhomme, 2004). The spatial patterns of genetic separation between the South (Faroe Islands, North Sea, Baltic Sea, Celtic Sea and Irish Sea) and the North (Nova Scotia and Newfoundland, Greenland, Iceland, Norway, Barents Sea, and White Sea) could be taken as evidence for this. The South is a shallow water environment whereas the North has more diversity of depth ranging from shallow to deep waters. Differences in temperature, salinity and other environmental factors are correlated with the North South difference. The great nucleotide divergence between the North and the South would imply either that this is an ancient divergence (not a Pleistocene event) or even a not-so-ancient divergence driven by strong selection over a shorter time. If the time of separation of G. morhua and G. macrocephalus and G. chalcogrammus is taken at 3.8–4.0 Mya (Coulson et al., 2006) the time of separation of the A and B clades would then be 2 Mya based on the nucleotide divergence of the A and B clades which we show is one half that of the closely related taxa. An even lower divergence time of 2.1 Mya has been suggested (Pogson & Mesa, 2004) that would still leave the divergence of the A and B clade at 1 Mya. These divergence times, however, are all based on the Kingman coalescent and the faster time scales of the multiple merger coalescent are discussed below.

A counter argument is that isolation and admixture are part of the breeding structure of a population leaving genome-wide impacts (Wright, 1931). Therefore, different genes should be concordant in their behavior (Bernardi, Sordino & Powers, 1993). This should be true for neutral genes that randomly drift apart in the different isolated areas. Genes under selection adapting to the different environments of the isolated areas should show even greater divergence. The HbA2 and the Myg show no differentiation between the North and the South. Also, the non-outlier SNPs in Moen et al. (2008) show no differentiation whereas three SNPs of the Ckma gene show high and extreme FST. The correspondence between our results and those of Moen et al. (2008), with very similar FST between our North vs. South and the north vs. south along the coast of Norway in Moen et al. (2008), is strong independent verification of our main result. The Ckma was, after all, the most extreme outlier in Moen et al. (2008). Similarly, Bradbury et al. (2010) found that non-outlier SNPs show no differentiation although other SNPs show differentiation from parallel adaptation to temperature on the eastern and western side of the Atlantic Ocean. Nielsen et al. (2003) described a pattern of microsatellite variation in a transition area between the Baltic and Danish Belt Sea which they interpret as a hybrid zone. There is no evidence for a hybrid zone at that location in the Ckma data. In fact, specific variants within the A allele are widely distributed among localities in the South including the Baltic Sea. This implies gene flow among localities in the South. Similar patterns within B alleles imply gene flow among localities in the North. If indeed there is a hybrid zone for the Ckma gene, it would lie between the Faroe Islands on one hand and Iceland and north and middle Norway on the other hand. Considering the North East Arctic and Coastal cod in Norway as an admixture of isolated populations (Pogson & Fevolden, 2003; Árnason & Pálsson, 1996) would add a third hybrid zone within the distribution of the species. It is not a parsimonious explanation to consider there to be multiple hybrid zones of secondary contact within distribution of the species.

For comparison one can consider the Pan I locus (Fevolden & Pogson, 1995; Fevolden & Pogson, 1997) that clearly is under selection (Pogson, 2001; Pogson & Mesa, 2004) related to depth and fisheries (Sarvas & Fevolden, 2005; Case et al., 2005; Árnason, Hernandez & Kristinsson, 2009). At face value, the locus shows similar differentiation between North and South (Sarvas & Fevolden, 2005) as the Ckma locus. However, the details differ and the parallels between the Pan I and Ckma genes are more apparent than real. Pogson & Fevolden (2003) argue that specific neutral alleles found within a functional class (the Pan I A allel) should show differences between historically isolated regions. Under the historical (isolation/admixture) hypothesis different neutral alleles will drift to high frequencies or fixation in geographic regions isolated from each other. Under the selection hypothesis, they should move seamlessly among localities within the putative isolated regions. Pogson & Fevolden (2003) tested the “historical” and “selection” hypotheses (c.f. Árnason & Pálsson, 1996) of Atlantic cod in northern Norway by studying presumed neutral variation among the PanI A alleles in coastal and Arctic localities. In short, they found no evidence supporting the historical hypothesis. In fact there were greater differences among coastal localities and between the two Arctic localites than overall between the Arctic and coastal areas. Because of the heterogeneity among coastal localities, Pogson & Fevolden (2003) also rejected the selection hypothesis because neutral mutations would move freely among localities within a region and should not show any structure. However, under a skewed offspring distribution and sweepstakes reproduction, there can be substanital population structure as measured by FST in the face of considerable gene flow (Eldon & Wakeley, 2009). Thus their results do not seem at odds with a multiple merger coalescent model.

For Pan I the B allele is largely absent in the South. But the absence of an allele from a certain region cannot be used as evidence for the isolation of populations from that region from populations in other regions. Instead under the isolation/admixture (historical) hypothesis one would expect (Pogson & Fevolden, 2003) specific Pan I A alleles to be present characterizing the South and another set of A alleles characterizing the North. But that is not the case; among the various A alleles there is no specific clade of Pan I A alleles in the South (U Hernandez & E Árnason, 2014, unpublished data). However, for the Ckma gene there is a specific allele, namely the A allele, that is at a high frequency and characterizes the South.

The Pan I B allele which is adapted to the deep (Pampoulie et al., 2007; Árnason, Hernandez & Kristinsson, 2009) is largely absent from the South. The Pan I B allele, which is found in the North and in deep water, is much less variable than the Pan I A alleles (Pogson, 2001). This is opposite to what we find for the Ckma A alleles (the South allele), which has less variation than the Ckma B allele (Fig. 1) although this is not seen in the summary statistics (Table 1) because of greater recombinational variation at the base of the A clade (Table S2). Also, the Pan I locus variation is more related to depth than to geography (Árnason, Hernandez & Kristinsson, 2009). Under the admixture hypothesis, these two loci and all loci showing genome wide effects are expected to show the same pattern.

Under the isolation/admixture hypothesis, one would expect recombinant types to be restricted geographically to the zone of secondary contact. This was not the case. Therefore, we think it is more likely that the two blocks of nucleotide sites are held together in linkage disequilibrium by epistatic fitness interactions and that there has been a build up of linkage disequilibrium over time.

Overall, therefore, we find that the Ckma gene does not fit the hypothesis of ancient divergence of gene pools and admixture in secondary contact.

Ξ colaescent and site frequency spectra

The trimodal site frequency spectrum is not predicted by any of the coalescent models considered here: the Kingman coalescent and the two Λ coalescent models, the Beta(2 − α, α) (Schweinsberg, 2003), and the point-mass coalescent (Eldon & Wakeley, 2006). Under the Λ coalescent, at most a single multiple merger event occurs at any one time. The distribution of family size is of interest, and the parameter α influences the probability of getting large families. Under the Beta(2 − α, α) coalescent model, the probability of a family size of k or more viable offspring decays like k−α (Schweinsberg, 2003) in the limit of a large k. The pool of viable offspring is then resampled to form the next generation under the same conditions. For the Kingman coalescent, α ≥ 2 and there is little chance of seeing large families. For the Beta(2 − α, α) coalescent, 1 ≤ α < 2 and the lower α the greater is the chance of seeing a large family (Schweinsberg, 2003). The ψ parameter of the point-mass coalescent (Eldon & Wakeley, 2006) similarly measures the proportion of the population that is the offspring of a single individual and is thus an indicator of reproductive sweepstakes. Our estimates of ψ indicate reproductive sweepstakes at the neutral loci and within the A and B alleles of Ckma. Balancing selection at Ckma lessens the effects of sweepstakes reproduction. Sweepstakes reproduction has been detected in other high-fecundity organisms (Hedgecock & Pudovkin, 2011; Harrang et al., 2013).

Under the more general Ξ coalescent 0 < α < 1 (Schweinsberg, 2000) there can be many large families independently in each generation. It would seem that this process could generate multimodal site frequency spectra. Indeed in simulations of Ξ coalescence site frequency spectra can display multiple modes (B Eldon, pers. commun., 2014). This question needs further theoretical work. In terms of the concept of sweepstakes reproduction, multiple local sweepstakes could have this effect on the site frequency spectrum. Under local sweepstakes, genetic structure may be ephemeral (Johnson & Wernham, 1999). Whether this affects the location of the modes and the exact shape of the site frequency spectrum under Ξ coalescent is not known. However, one would not expect build-up of sites around a specific mode of the site frequency spectrum or of two modes at opposite frequencies as at Ckma. Also, there should be no particular or regular geographical pattern. Therefore, we think that bumps in the site frequency spectrum under Ξ coalescent is not a good explanation for the Ckma spectrum.

Models of population growth can account for the high frequency of singletons. However, these models also do not predict the trimodal site frequency spectrum observed at Ckma. This is a locus specific behavior that is most likely due to balancing selection.

It is of course possible that population growth and sweepstakes could be occurring at the same time. We do not at this time have methods that estimate simultaneous multiple merger coalescents and population growth and evaluate the relative contribution of each. It is likely that disentangling the effects changes in population size and sweepstakes reproduction may be hard. For example, Birkner et al. (2009) discussed how recurrent bottlenecks may construct simultaneous multiple merger Ξ coalescent.

Balancing selection

Balancing selection generates long branches in the genealogy and neutral variation accumulates on the branches. The balanced functional types (the Ckma A and B alleles in this case) act as they were separate and isolated populations accumulating neutral variation. Recombination can bring variation from one branch to another acting like migration that brings alleles from one population to another (Charlesworth, Nordborg & Charlesworth, 1997; Charlesworth, Charlesworth & Barton, 2003; Charlesworth, 2006). However, the molecular signatures of balancing selection depend on many factors. Is it a long standing, even trans-species, polymorphism such as MHC in human and chimpanzee (Fan et al., 1989; Nei & Hughes, 1991) or is it very recent? Examples of the latter are human glucose 6 phosphate dehydrogenase (G6PD) (Verrelli et al., 2002), and hemoglobin β S (Currat et al., 2002) and hemoglobin β E (Ohashi et al., 2004) and spatially divergent selection of lactase persistence (Tishkoff et al., 2007; Ranciaro et al., 2014) in which a particular allele sweeps a chromosomal segment to an intermediate equilibrium frequency. In these instances, recombination has not had time to break up linkage disequilibrium, which can extend over large regions. There is very little variation among the new alleles while the alternative chromosomes show much more variation in this region representing the standing variation in the population at the start of the partial sweep.

The effects of a long standing single locus balancing selection will extend only short distances with free recombination and will be difficult to detect (Wiuf & Hein, 1999). If, however, there are obvious signs of a long standing balanced polymorphism it is likely due to a build-up of co-adapted complexes of epistatic interactions among multiple sites and/or suppression of recombination (Wiuf & Hein, 1999). The concept of a supergene of multiple co-adapted sites possibly locked together by structural variation (Thompson & Jiggins, 2014) such as found in butterfly mimicry (Joron et al., 2011) is relevant. There also can be both partial and complete selective sweeps of new types within each allele of a supergene. Such intra-allelic selective sweeps would reduce variation within and increase variation between alleles. Such reduction of variation could look similar to that for a recent balanced polymorphism, except that it would not be limited to one functional type. Thus Pogson (2001) argues that he has detected on-going partial sweeps within each of the two Pan I alleles of Atlantic cod.

Pogson & Mesa (2004) further argue that the Pan I polymorphism is older than speciation of Atlantic cod and Walleye pollock, the closest relatives. The Pan I locus is in a “genomic island” (Bradbury et al., 2013; Hemmer-Hansen et al., 2013) a potential supergene of co-adapted complexes possibly locked together by structural variation. U Hernandez & E Árnason (2014, unpublished data) find large number of differences between the two functional Pan I types in a 12.5 kb region around the PanI gene that are too extensive to be a partial sweep of a new allele. Such variation is likely to have built up over some time by selection (see time scales below). This is in face of considerable gene flow implied by lack of differentiation of neutral loci (Moen et al., 2009; Bradbury et al., 2010; Eiríksson & Árnason, 2013; Hemmer-Hansen et al., 2014). Similarly, the wide distribution of variants within both the A and B alleles of Ckma implies gene flow among localities within South and within North areas. The recombinant haplotypes between the A and B alleles of Ckma imply gene flow between the South and the North localities.

The observation that the amino acid sequences are conserved might be taken as evidence that there is only purifying selection at the locus. However, claiming balancing selection does not necessarily imply amino acid differences. There is evidence for positive selection in non-coding DNA in other systems (e.g., Drosophila, Andolfatto, 2005) and methods have been developed to detect positive and balancing selection in non-coding regions (e.g., Zhen & Andolfatto, 2012). Balancing selection has also been detected in regulatory regions in other systems. For example, the 5′ cis regulatory region of CCR5 shows evidence for balancing selection (Bamshad et al., 2002), as does the promoter region of the human Interleukin 10 gene (Wilson et al., 2006), a regulatory region upstream from the human UGT2B4 gene (Sun et al., 2011), the NE1 locus in modern Humans and Neanderthals (Gokcumen et al., 2013), and the 5′ UTR’s of upregulated genes and genes for effector proteins of a plant-pathogenic fungus (Rech et al., 2014). We have not identified a specific target of selection and we speculate that there is selection on regulatory regions (5′, 3′, intronic, and even silent sites that may influence regulation) of the Ckma gene.

Ckma had the highest FST among all loci studied by Moen et al. (2008) and, therefore, the focus of selection is likely either the gene itself or a very tightly linked locus. We have looked in www.ensemble.org what genes are in the close neighborhood. There are no obvious candidates among them for a gene under strong selection. However, we think that an answer to this question must await a more detailed analysis of a larger region around the Ckma gene.

The Kingman and multiple-merger Λ coalescent models that we apply here are models of neutrality. One could argue that it is not appropriate to apply such neutral models to the Ckma locus that is already suspected to be under selection. However, understanding how the locus deviates from neutrality is important for understanding the pattern of selection. Under the neutral theory (Kimura, 1983), polymorphism within species is the transient phase of molecular evolution that leads to divergence between species. This is the rational for the HKA test of selection or neutrality (Hudson, Kreitman & Aguadé, 1987) that neutrally evolving genomic regions should have the same proportion of polymorphism to divergence. Balancing selection would tend to increase the level of polymorphism within species relative to divergence between them. The results of HKA test are in the direction of balancing selection. The HKA test shows a relative slowing down of divergence to rate of polymorphism at the Ckma locus.

Similarly, we consider the peaks in the site frequency spectrum of the Ckma gene to be evidence for balancing selection. The trimodal site frequency spectrum with two high frequency peaks at opposite frequencies that fold into one peak in a folded site frequency spectrum points to the build-up of variation over time. Under a recent balanced polymorphism scenario, such as G6PD and β globins in humans, there would be one peak at a particular frequency in the site frequency spectrum representing all sites at which the new allele differs from the ancient alleles. There could be multiple peaks representing high frequency polymorphisms among the ancient alleles. However, they are not expected to be at opposite frequencies to the frequency of the new allele. Therefore, we argue that the pattern at Ckma represents a balanced polymorphism that has been built up over time.

Coalescent parameter estimates and time scales

The question of coalescent time scale, however, must be considered. Under the Kingman coalescent, time is measured in terms of N/σ2, with population size scaled by the variance of family size (Sagitov, 1999; Árnason, 2004; Tavaré, 2004). With a Poisson distribution of family size, σ2 = 1 for a constant size haploid population and, therefore, time scales with N under the Kingman coalescent. In an extreme winner-take-all sweepstakes, σ2 = N and a sample would coalesce in the previous generation and there would be no variation (Árnason, 2004). In more realistic multiple merger coalescent models, the time scale is the quantity cN=Eν1−12N−1 where cN is the probability of two lineages coalescing in the previous generation in a haploid population of fixed size N and ν1 is the random number of offspring of individual 1 (Sagitov, 1999). In general, the time scale of multiple merger coalescent models can be much shorter than for Kingman coalescent. Under the Beta(2 − α, α) coalescent model time scales with Nα−1 (Schweinsberg, 2003; Eldon et al., 2015). For this model, our estimates of α for the nuclear genes are quite low which implies very short time scales. The neutral genes would seem to coalesce in the very recent past. The A and B alleles of Ckma run on very similar time scales to the neutral genes and the locus itself at a slower rate due to the balancing selection with a time scale approximately the cube root of the effective population size Ne. The mitochondrial DNA runs at yet another and slower time scale. For mtDNA time scales with approximately the square root of N. Predicted turnover of alleles is faster and ages of alleles shorter under multiple merger coalescent (Eldon, 2012). Different populations and species may run on different time scales (Eldon & Degnan, 2012) complicating divergence time estimates. Estimates based on Kingman coalescent of divergence times of Atlantic cod populations (Bigg et al., 2008) or divergence of gadid taxa (Coulson et al., 2006) may therefore be too high and may need revision.

Conclusion

The Ckma protein coding sequence is conserved between all but the most distantly related Arctic cod. The amino acid variants are all singletons in the sample. Based on these facts, we conclude that the protein coding sequence is under purifying selection. At the same time, silent and non-coding variation at the locus shows extreme spatial differentiation with an FST greater than 0.8 between the North and the South regions. The regulatory function of this variation is unclear. We argue that the high and locus-specific FST, the highest seen so far for any locus and any spatial comparison in Atlantic cod, indicates that selection and not admixture of anciently divergent gene pools is responsible. Selection is likely to be very strong. It follows that Ckma (or an extremely tightly linked locus) is the focus of selection because the highest FST indicates the site of action of selection (Nielsen, 2005). Some of the variation may be neutral having risen in frequency within the balanced functional allele where it arose (Charlesworth, 2006). Alternatively, some of the variation may be due to selection building co-adapted complexes (Thompson & Jiggins, 2014). In addition to a high peak at singleton sites, higher than that predicted by the Kingman coalescent and characteristic of the multiple-merger coalescent, the site frequency spectrum has two high-frequency modes at opposite but matching frequencies representing the two branches of the genealogy. This pattern is further support for balancing selection. Our estimates of parameters of multiple-merger Λ coalescent show that time-scales are fast in accordance with theoretical expectations.

Supplemental Information

Figure S1 Map of sampling localities for Atlantic cod in the North Atlantic

Sampling localities in the waters of Canada (Nova Scotia and Newfoundland) Can, Greenland Gre, Iceland Ice, Norway Nor, Faroe Islands Far, and from the Barents Sea Bar, White Sea Whi, North Sea Nor, Baltic Sea Bal, Celtic Sea Cel, and Irish Sea Iri. Stippled parts indicate overall regions called North and South.

Click here for additional data file.

Figure S2 Structure of the Ckma gene and sequenced parts

Boxes represent exons, start (red), internal (magenta) and terminal (blue). Green boxes represent sequenced fragments trimmed to Phred score of at least 30. Up and down arrows mark TATA box and poly A signal starts respectively.

Click here for additional data file.

Figure S3 Maximum likelihood tree of creatin kinase proteins of paralogous genes in the Atlantic cod genome and Ckma orthologs from this study

Protein sequence of the two alleles A (Gmo.Nse A) and B (Gmo.Ice B) in Atlantic cod and of representatives from closely related taxa. Predicted protein isoforms from mitochondria (CKMT), brain (CKB) and muscle (CKM). Closely Related taxa are Boreogadus saida (Bsa), Gadus macrocephalus (Gma), Gadus ogac (Gog), and Gadus chalcogrammus (Gch).

Click here for additional data file.

Figure S4 Maximum likelihood tree of variation (A and B alleles) of Ckma among 122 Atlantic cod individuals

Color codes for Atlantic cod localities are the waters of Canada (Nova Scotia and Newfoundland) Can, Greenland Gre, Iceland Ice, Norway Nor, Faroe Islands Far, and from the Barents Sea Bar, White Sea Whi, North Sea Nor, Baltic Sea Bal, Celtic Sea Cel, and Irish Sea Iri. Stippled parts indicate overall regions called North and South.

Click here for additional data file.

Figure S5 Maximum likelihood tree of nucleotide variation of Myg among 45 Atlantic cod and two Gadus macrocephalus individuals

Color codes for species are Gadus macrocephalus Gma. Localities and color codes for Atlantic cod are the waters of Canada (Nova Scotia and Newfoundland) Can, Greenland Gre, Iceland Ice, Norway Nor, Faroe Islands Far, and from the Barents Sea Bar, White Sea Whi, North Sea Nor, Baltic Sea Bal, Celtic Sea Cel, and Irish Sea Iri.

Click here for additional data file.

Figure S6 Maximum likelihood tree of nucleotide variation of HbA2 gene among 113 Atlantic cod and 14 individuals of closely related taxa

Color codes for species are Boreogadus saida Bsa, Gadus chalcogramma Gch, Gadus macrocephalus Gma, and Gadus ogac Gog. Localities and color codes for Atlantic cod are the waters of Canada (Nova Scotia and Newfoundland) Can, Greenland Gre, Iceland Ice, Norway Nor, Faroe Islands Far, and from the Barents Sea Bar, White Sea Whi, North Sea Nor, Baltic Sea Bal, Celtic Sea Cel, and Irish Sea Iri.

Click here for additional data file.

Figure S7 Linkage disequilibrium heatmap of D’ among intermediate frequency polymorphic sites for a 2500 bp fragment of the Ckma gene of Atlantic cod

The approximate locations of sequenced parts and exon intron structure are the same as in Fig. S2.

Click here for additional data file.

Figure S8 Linkage disequilibrium heatmap of D’ of a 2500 bp fragment of the Ckma gene of Atlantic cod

Singleton variable sites were excluded from the analysis. The approximate locations of sequenced parts and exon intron structure are the same as in Fig. S2.

Click here for additional data file.

Figure S9 Unfolded site frequency spectrum of Atlantic cod HbA2 gene

Gadus macrocephalus was used as the outgroup. Theroretical expectation under Kingman coalescent (red dots), Beta(2 − α, α) coalescent (magenta squares), and point-mass coalescent (blue stars).

Click here for additional data file.

Figure S10 Unfolded site frequency spectrum of Atlantic cod Myg gene

Gadus macrocephalus was used as the outgroup. Theroretical expectation under Kingman coalescent (red dots), Beta(2 − α, α) coalescent (magenta squares), and point-mass coalescent (blue stars).

Click here for additional data file.

Figure S11 Unfolded site frequency spectrum of Atlantic cod Ckma A alleles (A) and B alleles (B)

Gadus macrocephalus was used as the outgroup. Number of individuals n = 43 and n = 79 respectively. Theroretical expectation under Kingman coalescent (red dots), Beta(2 − α, α) coalescent (magenta squares), and point-mass coalescent (blue stars).

Click here for additional data file.

Figure S12 The ℓ2 distance for the unfolded site frequency spectrum of the nuclear genes Myg, Hb2A, Ckma, and the Ckma-A and Ckma-B alleles of Ckma on the α parameter of the Beta(2 − α, α) coalescent (A)

Click here for additional data file.

Figure S13 The ℓ2 distance for the unfolded site frequency spectrum of mtDNA from various localities of the North Atlantic on the α parameter of the Beta(2 − α, α) coalescent (A) and the ψ parameter of the point-mass coalescent (B)

Click here for additional data file.

Table S1 Primer sequences for amplification from Atlantic cod and closely related taxa and sequencing fragmentsof Ckma gene

Click here for additional data file.

Table S2 Segregating sites of the Ckma gene among 122 Atlantic cod individuals and 10 individuals of closely related species

The top panel of the boxhead indicates 5′ UTR, exon, intron, and 3′ UTR location of segregating sites, numbers represent exon and intron numbers, s, n, -, u, and ∇ represent synonymous, nonsynonymous, intronic, UTR, and indel sites respectively. The middle panel of the boxhead are site numbers of the Ckma gene (3,604 bp) and the bottom panel are site numbers of the concatenated 2,500 bp sequenced fragment (Fig. S2). Horizontal lines are drawn between A and B alleles and between Atlantic cod and closely related species Gma Gadus macrocephalus, Gch Gadus chalcogrammus, Gog Gadus ogac, and Bsa Boreogadus saida. Locality codes are as indicated in methods.

∇1 is a four bp indel at sites 573–576. ∇2 is an 11 bp indel at sites 941–951. ∇3 is a two bp indel at sites 1,470–1,471. ∇4 is an 14 bp indel at sites 1,828–1,841. ∇5 is an three bp indel at sites 2,127–2,129. ∇6 is a one to eight bp indel at sites 2,217–2,224. ∇7 is a one bp indel at site 2,354. ∇8 is a one bp indel at site 2,476.

The Moen et al.Gm366-0514 locus with an FST = 0.83 is at site 1,634, the Gm366-1022 locus with an FST = 0.82 is at site 2,426, and the Gm366-1073 with an FST = 0.82 is at site 2,476. A alleles are in red and B alleles in blue. Closely related species are colored: Gma is green, Gog is darkgreen, Gch is darkred, and Bsa is darkblue.

Click here for additional data file.

Table S3 Gross Dxy and net Da nucleotide divergence per site between Gadus morhua Gmo and Gadus macrocephalus Gma and Gadus chalcogrammus Gch and between A and B alleles of Atlantic cod

Divergence, D, and standard deviation, s, found using Jukes and Cantor correction.

Click here for additional data file.

Table S4 Non-synonymous changes within and between species

Individuals with species and locality codes, first aa represents majority and the second the change, position refers to position in concatenated sequence in Table S2.

Click here for additional data file.

Table S5 Maximum likelihood analysis of a Kingman-coalescent HKA test of neutrality and selection at three genes in Atlantic cod

Test is twice the ln L difference of the two models, neutrality and selection at Ckma. Three loci are under test: Hemoglobin α2 (Hbα2), Myoglobin (Myg), and Creatine Kinase Muscle (Ckma). θ is the scaled effective population size and the parameter k measures changes in diversity due to selection. Based on method of Wright and Charlesworth (2004).

Click here for additional data file.

Table S6 Frequency of A and B alleles in different localities

Click here for additional data file.

Table S7 Pairwise FST values (lower triangular) of population differentiation among localities

Probabilities in black on upper triangular, boldface are significant P values. North (blue) and South (red) defined ad hoc by results.

Click here for additional data file.

Table S8 Likelihood ratio test statistics G for observed site frequency spectra of Ckma and expectation according to different coalescent models

Click here for additional data file.

Table S9 The ℓ2 distance (Birkner et al., 2013b) between the observed site frequency spectra and expectation according to the Beta(2 − α, α) and point-mass multiple merger coalescent models for the three genes, Ckma, HbA2, and Myg

Click here for additional data file.

Table S10 The ℓ2 distance and approximate log Likelihood (Eldon et al., 2015) between the observed site frequency spectra and expectation according to the algebraic (A, γ) and exponential (E, β) growth models (Eldon et al., 2015) for the three genes, Ckma, HbA2, and Myg.

Click here for additional data file.

We would like to thank Jarle Mork (Norwegian University of Science and Technology), Kristján Kristjánsson (Marine Research Institute in Reykjavik), Grant Pogson (University of California at Santa Cruz), Remment ter Hofstede (Institute for Marine Resources and Ecosystem Studies in the Netherlands), and Michael Canino (National Oceanic and Atmospheric Administration) for help in securing some of the samples. We would like to thank Brenda Ciervo Adarna, Guðni Magnús Eiríksson, Lilja Stefánsdóttir, Ragnheiður Fossdal, Svava Ingimarsdóttir, Ubaldo Benitez Hernandez for help with some of the laboratory work. We would like to thank Bjarki Eldon for programs and help with coalescent parameter estimation and for critical comments on the manuscript. We would like to thank R.C. Lewontin for discussions and critical comments on the manuscript. We would also like to thank reviewer Michael Matschiner and an anonymous reviewer for critical comments that helped us improve the paper.

Additional Information and Declarations

Competing Interests

Author Contributions

Animal Ethics

DNA Deposition

Data Deposition

The authors declare there are no competing interests.

Einar Árnason conceived and designed the experiments, performed the experiments, analyzed the data, contributed reagents/materials/analysis tools, wrote the paper, prepared figures and/or tables, reviewed drafts of the paper.

Katrín Halldórsdóttir conceived and designed the experiments, performed the experiments, analyzed the data, wrote the paper, reviewed drafts of the paper, submitted sequences to GenBank.

The following information was supplied relating to ethical approvals (i.e., approving body and any reference numbers):

The Icelandic Committee for Welfare of Experimental Animals,

Chief Veterinary Office at the Ministry of Agriculture,

Reykjavik, Iceland has determined that the research conducted

here is not subject to the laws concerning the Welfare of

Experimental Animals (The Icelandic Law on Animal Protection,

Law 15/1994, last updated with Law 157/2012). DNA was isolated

from tissue taken from dead fish on board research vessels. Fish

were collected during the yearly surveys of the Icelandic Marine

Research Institute. All research plans and sampling of fish,

including the ones for the current project, have been evaluated

and approved by the Marine Research Institute Board of

Directors. The Board comprises the Director General, Deputy

Directors for Science and Finance and heads of the Marine

Environment Section, the Marine Resources Section, and the

Fisheries Advisory Section. Samples were also obtained from dead

fish from marine research institutes in Norway, the Netherlands,

Canada and the US that were similarly approved by the respective

ethics boards. The samples from the US used in this study have

been described in Cunningham et al. (2009) and the samples from

Norway in Árnason & Pálsson (1996). The samples from Canada

consisted of DNA isolated from the samples described in

Pogson (2001). The samples from the Netherlands were obtained

from the Beam-Trawl-Survey

(http://www.wageningenur.nl/en/Expertise-Services/

Research-Institutes/imares/Weblogs/Beam-Trawl-Survey.htm)

of the Institute for Marine Resources & Ecosystem Studies (IMARES), Wageningen

University, the Netherlands, which is approved by the IMARES Animal Care Committee

and IMARES Board of Directors.

The following information was supplied regarding the deposition of DNA sequences:

GenBank

KM624178–KM624309

KM624310–KM624436

KM624437–KM624483.

The following information was supplied regarding the deposition of related data:

DataDryad

http://datadryad.org/.

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
