# Peer review of "Nucleotide variation and balancing selection at the Ckma gene in Atlantic cod: analysis with multiple merger coalescent models"

_PeerJ, doi:10.7717/peerj.786_

## Round 0.1 · original submission · Major Revisions

The manuscript has been seen by two reviewers who have assessed the paper in some detail and who both provide many useful comments. Importantly, reviewer #2 raises conceptual concern regarding the interpretation of balancing selection and (absence of) admixture, as well as regarding methodology. This will require extensive revision and re-review.

·

Basic reporting

Reporting is comprehensive and most parts are relatively easy to follow even though a rather broad theoretical and empirical background literature is referred to. The background is largely discussed adequately and references are placed appropriately.

The manuscript could benefit from a more basic explanation, why multiple merger coalescents are expected to perform better than the Kingman coalescent with high-fecundity species. I agree that with high-fecundity species, the basic assumption of similar numbers of offspring between individuals is violated, however, it is not entirely clear why this should have a strong effect when empirical data sets are used, especially on a large geographic scale, as in the case of this study. After all, we would not expect samples from Greenland and Norway to share the same parents, and even on a more local scale, I would expect few if any of the sampled individuals to be siblings originating from a single high-fecundity individual. Nevertheless, the statistical comparisons do in fact support the use of multiple merger coalescents. Why is this so?

The use of the multiple merger coalescent with empirical data may most interesting for many readers, but little information is given on its practical application. The authors write that they use "analyzed site frequency sprectra using ... statistical methods developed for multiple merger Lambda coalescents (Birkner et al., 2013b)", but it is not specified whether code provided by Birkner et al. was used or whether calculations were performed with code developed by the authors. Also, if there are settings to be chosen for either code, these details would be very helpful.

The structure of the discussion could be improved. It's not entirely clear what the three possible scenarios (l. 356) to explain allelic divergence are.

I must say that I find the phrasing partially clumsy, and have highlighted the most obvious examples in the attached annotated manuscript with comments (written with Apple Preview, if these should not be readable for you, please get back to me). I would appreciate if the authors could once more go through their wording, especially for the highlighted sentences but also in other parts of the manuscript.

Supplementary S2 was not available for review, but does not seem to be crucial to assess the quality of the manuscript. Nevertheless this information should be made available to the reader.

Experimental design

The experimental design is interesting and sound. Some more details should be given regarding the application of the multiple merger coalescent (see above).

Validity of the findings

All findings are valid and discussed appropriately.

Reviewer 2 ·

Basic reporting

This is an interesting manuscript that investigates the patterns of nucleotide sequence polymorphism at the Cmka gene in Atlantic cod populations sampled across the north Atlantic region. The Cmka locus was previously identified as an FST-outlier in the NE Atlantic region by Moen et al. (2008). The motivation behind the present study is admirable – too often adaptive stories are made around the discovery of FST-outliers without the necessary follow-up studies to produce evidence for the action of natural selection. The objective of the present study was to assess the evidence for natural selection at this FST-outlier by comparing the patterns of nucleotide sequence variation at Cmka to two randomly chosen neutral genes (HbA2 and Myg). Unlike the neutral loci, two long-lived alleles were detected at Cmka that the authors interpret as evidence for balancing selection. Multiple merger coalescent models were used to evaluate the support for sweepstakes recruitment.

Experimental design

This manuscript is well written, well organized, and methodologically sound. The application of multiple merger coalescent models is innovative and novel.

Validity of the findings

I have a number of concerns with the interpretation of the results and their relation to previous studies on cod.

First, there is no evidence for balancing selection at Cmka. No amino acid differences were found between the Cmka A and B alleles that could account for their long persistence in cod populations. This seriously undermines the argument that the Cmka locus is experiencing balancing selection. The authors suggest that some of the silent or non-coding mutations might be influencing expression of the gene in different tissues or under different environmental conditions (lines 345-347) but this strikes me as overly speculative. The long-term residency of Cmka alleles in the absence of any functional changes is difficult to explain except under the admixture hypothesis (discussed below). The Cmka gene must reside in a region where selection is targeting a closely-linked gene. It would be interesting to see what other genes are in this chromosomal region.

Second, the authors reject the admixture hypothesis for their Cmka data by showing how it contrasts with the pantophysin gene. However, the parallels between these two genes are so striking that the historical isolation hypothesis becomes much stronger on the basis of the Cmka data. The gene genealogies of PanI and Cmka are remarkably similar. Both have two old alleles with comparable numbers of mutational differences but the branches separating PanI alleles are longer due to the repeated episodes of positive selection. One interesting difference between the PanI and Cmka genealogies is the more extensive recombination between allelic groups at Cmka (Figure 2). In the absence of any functional differences between Cmka alleles, it is difficult to explain how recombination would not have eroded all of the divergence between the allelic groups unless they spent most of their existence in allopatry. It would be nice to a more detailed treatment of recombinant alleles (especially those at the very bottom of the genealogy shown in of Figure 2). The patterns of recombination might provide clues about when the putative admixture event occurred.

On lines 408-410, the authors reject the admixture hypothesis because they claim it predicts that different genes across the genome should show similar patterns. This prediction is true for neutral genes but not for genes experiencing selection like PanI. In fact, selection acting on recently-derived allelic variants at pantophysin that can account for the different patterns of variation between it and Cmka. For example, the PanI B allele in northern Norway is recently derived from an older B lineage that is restricted to the NW Atlantic. This allelic group was called the delta 2 B alleles by Pogson (2001) because of a 12 base pair intron deletion that disrupted a stem-loop structure in the second intron. The delta 2 B alleles display almost no polymorphism yet exist at high frequencies in northern Norway. This is suggestive of a recent ongoing selective sweep and explains why the PanI B allelic group in the far NE Atlantic (low polymorphism) shows a different pattern than the Cmka alleles in this area (high polymorphism). To claim that this pattern is not consistent with the admixture hypothesis is thus incorrect. It is incorrect because it doesn’t take into account the selection occurring at PanI.

Third, the data presented in Table S6 leads me to question whether the Cmka primers have successfully amplified all A and B alleles present across the north Atlantic. What population samples were used to create the PCR primers? Is there a possibility that subgroups of Cmka A and B alleles exist that could not be amplified from some individuals? After all, the great age of the Cmka polymorphism was unexpected and thus the primers weren’t constructed to necessarily deal with this situation. The complete absence of A alleles in the NW Atlantic and B alleles in the southern NE Atlantic is remarkable and contradicts the results from all previous population genetic surveys on cod over the past 50 years. Large differences in allele frequencies have been described between populations on either side of the north Atlantic, but I am not aware of any case where alleles are completely missing. Perhaps this is a consequence of the small sample sizes of alleles sequenced per locality in the present study. The correspondence between the high FST SNPs identified by Moen et al. (2008) and those observed in the present study is reassuring, but this does not mean that ascertainment bias could be responsible for alleles being missed from the NW Atlantic. The small numbers of alleles sequenced prevents robust tests for HWE. In this regard, a better strategy may have been to first estimate Cmka allele frequencies in larger samples and then take subsamples of these for sequencing. Conformity to HWE would also have provided strong evidence that a single Cmka gene was being amplified and that all alleles were being amplified and sequenced.

Fourth, are the applications of the various coalescent models appropriate if the Cmka gene is experiencing selection? The observed trimodal site frequency spectrum at Cmka is very unsual and clearly a consequence of the long persistence of the A and B alleles. What model would be appropriate to use in this instance? It seems that under any conceivable scenario (i.e., historical isolation or long-lived balancing selection) even the multiple merger models are inappropriate. It is not clear what is being learned here.

Fifth, can’t the observed site frequency spectra in cod be produced by a large recent expansion of the population following the last glacial cycle? Wouldn’t rapid population growth produce a large excess of low frequency alleles similar to that that expected under a sweepstakes recruitment scenario. Couldn’t both population expansion and sweepstakes recruitment be occurring together? Is there a way to distinguish between or evaluate the relative contributions of these hypotheses?

Additional comments

Some specific comments:

Lines 62-68. The human and cod studies are not directly comparable. The Moen et al. (2008) study used EST-derived SNPs, which have a much higher probability of being in linkage disequilibrium with selected mutations within and near protein-coding genes.

Lines 100-104. The study of a well-defined and well-understood gene is certainly a good starting point. However, the selective agents acting on the gene are usually completely unknown. This is often a greater limitation than not knowing much about the gene under study.

Line 118. “Walleye” is misspelled.

Line 171. “Statistical Analyses”?

Table 1. Perhaps the samples corresponding to the North and South could be given here below the Table?

---

## Round 0.2 · accepted · Accept

The authors have addressed the reviewers' comments in a reasonable way. In several cases they do not agree with reviewers' comments and they provide an explanation for that.